Modeling target-density-based cull strategies to contain foot-and-mouth disease outbreaks

http://orcid.org/0000-0002-1125-7210 Seibel Rachel L. 1 2
Meadows Amanda J. 2 3
Mundt Christopher 2
Tildesley Michael 1 M.J.Tildesley@warwick.ac.uk
1 Mathematics Institute, University of Warwick , Coventry, West Midlands , United Kingdom
2 Department of Botany and Plant Pathology, Oregon State University , Corvallis, OR , United States
3 Ginkgo Bioworks , San Bruno, California , United States
Ndeffo Martial
Electronic publication date: 2024 Feb 29
Publication date: 2024
Volume: 12
Electronic Location ID: e16998
Received 2023 May 18; Accepted 2024 Feb 2
Copyright: © 2024 Seibel et al.
Copyright year: 2024
Copyright holder: Seibel et al.
License: This is an open access article distributed under the terms of the Creative Commons Attribution License, which permits unrestricted use, distribution, reproduction and adaptation in any medium and for any purpose provided that it is properly attributed. For attribution, the original author(s), title, publication source (PeerJ) and either DOI or URL of the article must be cited.
License URL: https://creativecommons.org/licenses/by/4.0/

Keywords: Disease modeling, Emerging infectious diseases, Host density, Farm demography, Foot-and-mouth disease, Livestock disease, Ring culling

Funding: USDA National Institute of Food and Agriculture 2015-67013-23818 and 2022-67015-38059 NSF/NIH/USDA/BBSRC/BSF Ecology and Evolution of Infectious Diseases Program This project was funded by the USDA National Institute of Food and Agriculture (Award numbers 2015-67013-23818 and 2022-67015-38059) via the NSF/NIH/USDA/BBSRC/BSF Ecology and Evolution of Infectious Diseases Program. There was no additional external funding received for this study. The funders had no role in study design, data collection and analysis, decision to publish, or preparation of the manuscript.

==============================
Total ring depopulation is sometimes used as a management strategy for emerging infectious diseases in livestock, which raises ethical concerns regarding the potential slaughter of large numbers of healthy animals. We evaluated a farm-density-based ring culling strategy to control foot-and-mouth disease (FMD) in the United Kingdom (UK), which may allow for some farms within rings around infected premises (IPs) to escape depopulation. We simulated this reduced farm density, or “target density”, strategy using a spatially-explicit, stochastic, state-transition algorithm. We modeled FMD spread in four counties in the UK that have different farm demographics, using 740,000 simulations in a full-factorial analysis of epidemic impact measures (i.e., culled animals, culled farms, and epidemic length) and cull strategy parameters (i.e., target farm density, daily farm cull capacity, and cull radius). All of the cull strategy parameters listed above were drivers of epidemic impact. Our simulated target density strategy was usually more effective at combatting FMD compared with traditional total ring depopulation when considering mean culled animals and culled farms and was especially effective when daily farm cull capacity was low. The differences in epidemic impact measures among the counties are likely driven by farm demography, especially differences in cattle and farm density. To prevent over-culling and the associated economic, organizational, ethical, and psychological impacts, the target density strategy may be worth considering in decision-making processes for future control of FMD and other diseases.

Introduction

During infectious disease outbreaks, disease managers strive to deploy limited resources to optimize the impact of control given resource constraints. Mathematical models can serve as useful tools to help determine the best distribution of these limited resources by providing a means to simulate mitigation strategies and compare epidemic measures such as epidemic size, outbreak duration, and economic impact.

Host density is often an important determinant of disease spread, and a variable that can be manipulated for epidemic mitigation. Reducing host population density through culling has been used to suppress epizootics of both wild and domesticated animals (Miguel et al., 2020). In the context of highly contagious livestock diseases such as foot-and-mouth disease (FMD), mitigation strategies include quarantining, reactive culling (killing of animals on infected premises), and disinfecting infected premises, but these measures may be combined with preemptive culling (Gibbens et al., 2001; Bouma et al., 2003) or reactive vaccination (Muroga et al., 2012; Park et al., 2013) in more severe outbreaks. Preemptive control measures (e.g., livestock movement bans, culling of livestock neighboring premises, preemptive vaccination, etc.) were instrumental in containing previous FMD outbreaks occurring in livestock-dense regions in the UK (Tildesley et al., 2009), the Netherlands (Pluimers et al., 2002), and Japan (Muroga et al., 2012).

One of the most severe ways preemptive culls have been enacted to curb FMD transmission is through ‘ring culling’ (Ferguson, Donnelly & Anderson, 2001a), which aims to cull all susceptible livestock located on farms within a specified cull radius of infected premises (IPs). This ‘total ring depopulation’ approach was utilized in farming regions that were severely impacted by FMD to slow disease transmission by reducing the density of susceptible hosts and by removing potentially undiagnosed infectious hosts (Tildesley et al., 2009). During the UK 2001 epidemic, ring culling was not implemented in all counties (primarily implemented in Dumfries, Galloway, and Cumbria) and it was often targeted at sheep. Although ring culling can be effective at curbing livestock disease outbreaks, it can result in large numbers of livestock culled and sometimes overshoots the number of farms that need to be culled to stop transmission (Tildesley et al., 2009).

Vaccination may provide an alternative to a total ring depopulation preemptive cull policy, but it comes with complications as well. There are two methods of reactive ring vaccination: ‘vaccination-to-cull’ and ‘vaccination-to-live.’ The former involves vaccinating animals as quickly as possible to limit FMD transmission and subsequent culling of all vaccinated animals. The vaccination-to-cull strategy was used to control FMD in the Netherlands 2001 outbreak (Bouma et al., 2003). Vaccination-to-live involves vaccinating animals as a protective measure and does not lead to culling. Though the protective vaccination strategy is more ethical and less expensive to implement, there also are potential challenges associated with variation in vaccine efficacy depending on the antigenic match to the virus in circulation (Lyons et al., 2016), the waiting period of 4–5 days before the vaccine could prevent disease in vaccinated animals (Barnett & Carabin, 2002), the additional waiting time required to regain ‘FMD free’ status and access to international trade (OIE World Organisation for Animal Health, 2017), animals still spreading infection after vaccination (Roth, 2004), unreliable detection of these infected animals (Roth, 2004), and export restrictions as a barrier to implementation (Porphyre et al., 2013).

Here, we use a previously developed spatial FMD model (Keeling et al., 2001) to simulate hypothetical FMD outbreaks and control strategies in regions similar in area to Cumbria, UK, which was one of the epidemic hotspots during the 2001 epidemic, and to three other counties with differing farm demography. We aimed to explore the possibility of refining the total ring depopulation policy to uncover alternative cull-based control strategies that provide protection similar to the total ring depopulation policy without as many control casualties. We modified previously-utilized preemptive ring culling strategies by loosening the scorched earth policy and allowing for a low density of farms to remain within the cull radius. This means that not all farms within the cull radius are culled. This strategy aims to lower the density to a ‘target density’ of farms surrounding confirmed FMD cases (IPs) without eliminating farms in the cull radius. Target density is defined as the farm density within each ring around IPs after culling. We compared control strategy parameters including the target density, cull capacity (the number of farms per county that can be culled per day), and cull radius and their relative effects on epidemic severity by comparing traditional total ring depopulation ring culling to a target density-based ring culling strategy. It is important to consider that there may be additional factors at play in disease epidemics, including limitations in finances, human resources, and personal decisions to report disease.

Methods

Model

The model used in this article is a spatially-explicit, stochastic, state-transition simulation model first developed by Keeling et al. (2001) to predict FMD spread during the 2001 epidemic in the United Kingdom (UK). It has since been refined (Tildesley et al., 2008) and adapted to model FMD outbreaks in the United States and elsewhere (Tildesley et al., 2010; Tildesley, Smith & Keeling, 2012; Meadows et al., 2018). The fitting of this model to the 2001 UK FMD epidemic was done using Approximate Bayesian Computation and has been explored extensively in terms of investigating how well the model fits to the observed outbreak in 2001 (Tildesley & Keeling, 2008). The model has further been used to study potential impacts of spatial clustering (Tildesley et al., 2010), aggregated landcover data (Tildesley, Smith & Keeling, 2012), and real-time decision making (Probert et al., 2018).

As FMD spreads rapidly within farms, the model treats the farm as the unit of infection, classifying all animals on the same premises as susceptible, infected, infectious, or culled. In this study, we define ‘animals’ and ‘livestock’ as including cattle and sheep as there were many fewer pigs infected than cattle and sheep during the 2001 outbreak in the UK. The latent period is the time between when a farm becomes infected and when it becomes infectious and was set to 5 days (Tildesley & Keeling, 2008) in this study. After this period, the farm remains infectious for several days before being reported and culled, which was assumed to be 7 days in our simulations. Our simulations are based on data collected after the ban on livestock movement was implemented. We thus assume no livestock movement between farms and the model does not allow for new hosts to enter the system.

The biological core of the model is relatively simple and consists of two processes: calculation of the susceptibility/transmissibility of each farm and the dispersal kernel that describes transmission among farms based on contact tracing during the 2001 epidemic in the UK (Keeling et al., 2001). The kernel is thus based on all modes of transmission (direct, indirect, and airborne). However, contact tracing was conducted after implementation of the livestock movement ban and thus is influenced little by inter-farm transport of animals. The risk of infection between any two farms is determined by the number of livestock on infected and susceptible farms and the Euclidean distance between the farms. The rate of transmission between infectious farm i and susceptible farm j is given by:

Rateij=(TcowNcow,iq+TsheepNsheep,iq)×(ScowNcow,jp+SsheepNsheep,jp)×K(r)

Ns is the number of livestock species s on a farm, Ss and Ts respectively measure the species-specific susceptibility and transmissibility (Keeling et al., 2001), which scale non-linearly with the number of animals on a farm using parameters p and q (Tildesley et al., 2008). K is the distance-dependent transmission kernel, which models the decrease in risk of infection as the distance between infectious farm i and susceptible farm j (r) increases. We used a modified power-law transmission kernel:

K(r)=ζc+rαθ

θ (kernel width) is set to 1, α (kernel shape) is 2.2, ζ (kernel height) is set to 0.02, c (kernel offset) is set to 0.1, which are within the range of values used in previous FMD modeling studies (Chis Ster & Ferguson, 2007; Meadows et al., 2018).

Transmission of disease among all farms in a given county is calculated on a daily basis. The mean probability of a given farm becoming infected each day is the sum of calculated transmissions from all other farms on that day. The model uses a Monte Carlo simulation approach to evaluate potential epidemic outcomes. Whether each farm actually becomes infected each day is determined by drawing a random value from a distribution with a mean equal to mean probability of a given farm becoming infected each day. The disease status of each farm (susceptible, infected, infectious, and culled) is updated daily. Multiple runs of the simulation, usually 1,000 in this study, are then used to evaluate variability of outcomes under the influence of stochasticity.

County selection

Four counties in the United Kingdom (UK) were selected for applying this stochastic simulation-based model (Fig. 1). During the UK FMD epidemic in 2001, Cumbria was one of the most affected regions, with 2,961 farms (total of IPs and neighboring farms) being culled. Since there was a large spread of disease in this county, we selected Cumbria as the first county to explore simulated epidemics. Aberdeenshire was minimally impacted by the 2001 epidemic, with only one farm (an IP) being culled. Compared with Cumbria, Aberdeenshire had a substantially lower value for all demographic statistics, except that it had a slightly higher number of cattle per farm (Table 1). Devon and North Yorkshire were similarly impacted in the 2001 epidemic, and were intermediate between Cumbria and Aberdeenshire. The number of farms culled (total of IPs and neighboring farms) was 776 in Devon and 636 in North Yorkshire. Devon had a larger number of farms, but lower numbers of cattle and sheep per farm than Cumbria (Table 1). North Yorkshire had a similar number of farms to Cumbria, but it had slightly fewer animals per farm (Table 1). Each county varied in mean farm density (farms/km2), with Devon having the highest mean farm density of the four counties (Table 1, Fig. 2). To investigate the effects of farm demography on epidemic impact measures (animals culled, farms culled, and epidemic length), we selected these four counties given their differing farm demographics (farm density, species composition, and farm size) and differing epidemic intensities in 2001.

Figure 1 Map of counties in the United Kingdom in 2001.

The shaded polygons indicate counties that were used in this study (Aberdeenshire, Cumbria, Devon, and North Yorkshire). The different colors indicate different counties. The county names are labeled next to each shaded region. Source: Office for National Statistics licensed under the Open Government Licence v.3. Contains OS data © Crown copyright and database right (2023).

Table 1 Farm demography measures for Aberdeenshire, Cumbria, Devon, and North Yorkshire counties of the United Kingdom.

Measurement	Aberdeenshire	Cumbria	Devon	North Yorkshire	
Total farms	2,925	7,884	10,656	7,599	
Total area (km2)	6,312	6,768	6,708	8,653	
Farms/km2	0.46	1.16	1.59	0.88	
Total animals	819,593	3,892,268	2,954,237	3,012,423	
Total cattle	279,278	587,918	695,655	457,899	
Total sheep	540,315	3,304,350	2,258,582	2,554,524	
Animals/farm	280 ± 554	494 ± 757	277 ± 483	396 ± 638	
Cattle/farm	95 ± 139	75 ± 106	65 ± 98	60 ± 107	
Sheep/farm	185 ± 553	419 ± 723	212 ± 451	336 ± 608	
Animals/km2	130	575	440	348	
Cattle/km2	44	87	104	53	
Sheep/km2	86	488	337	295	
Note:

The measurements for animals (cattle + sheep), cattle, and sheep per farm are expressed in mean ± standard deviation.

Figure 2 Mean farm density (farms/km2) against radius around each farm (km) for Aberdeenshire, Cumbria, Devon, and North Yorkshire in the UK.

The x-axis starts at 1 km.

Defining target density

Ring culling is one strategy that has been implemented to combat FMD and involves culling farms within a specified ring size around IPs. In the case of total ring depopulation, all farms may be targeted for culling within the indicated areas, though in some cases such a policy may be directed at farms with a particular species (Kitching, Hutber & Thrusfield, 2004; Thrusfield et al., 2005; Kitching, Thrusfield & Taylor, 2006). We explored an intervention strategy that involves culling farms until a target density is reached within each ring. For each ring, the greater the target density the fewer farms are culled. All cattle and sheep on a selected farm are culled, as was the case for culled farms in the 2001 outbreak. The largest farm, in terms of overall head of livestock (cattle and sheep), is selected for culling, followed by farms with decreasing head of livestock, until the target density is reached. For example, suppose a designated ring around an IP of 1 km2 (radius=1πkm) contains 4 farms, including the IP. In the case of a total ring depopulation strategy, all 4 farms within the ring are culled. If the target density is 2 farms/km2, then the IP, as well as the largest farm (excluding the IP from selection), will be culled. We will later discuss other selection strategies for future consideration.

Control strategies

We ran a factorial analysis of intervention parameters and epidemic impact measures with the following control strategy parameters: target density, cull capacity, and cull radius. The target density strategy means that all farms falling within the cull radius around infected premises (IPs) were identified, but we only cull enough farms to reach the respective farm density within the cull radius. This culling is done reactively. For instance, if a neighboring farm has already been infected and farms in the ring have already been culled, then farms that have already been culled are not considered in the density calculation. Culls were conducted on a single day. If targeted farms on a given day exceed cull capacity, they are added to a queue. IPs are immediately put to the top of the list, but the others are culled as the daily capacity allows. The target density in traditional total ring depopulation ring culling policies would be 0 farms/km2; all farms within control radii are identified and culled. The purpose of this target density method is to identify an optimal farm density to achieve in control regions that is similarly effective at slowing the spread of disease as total ring depopulation but results in lower epidemic impact measures (animals culled, farms culled, and epidemic length) due to a reduction in control-based culls. Resources are often limited during outbreaks so it is important to explore situations in which resources cannot be allocated to all farms within control radii. Cull radius is the radius around IPs that identifies which farms are to be culled. Larger control radii generally lead to more farms culled per ring. By running a factorial by county, target density, cull capacity, and cull radius, we can inform ring-cull-based disease management strategies for Aberdeenshire, Cumbria, Devon and North Yorkshire in the UK.

Simulation details

We evaluated control strategies in simulations performed using cattle and sheep farm demography from the counties Aberdeenshire, Cumbria, Devon and North Yorkshire in the UK in 2001 (Table 1). Models were run independently for each county and thus did not include regional or national dynamics. We selected six target densities (0.0, 0.05, 0.1, 0.15, 0.2, and 0.4 farms/km2), five daily cull capacities (5, 10, 20, 100, and total farms), and seven control radii (0, 0.5, 1, 2, 3, 4, and 5 km). A cull radius of 0 km was simulated using only a target density of 0 farms/km2 since only the IP is culled in this case. In total, we simulated 185 unique control scenarios for each county. Total farms refer to the total number of farms in each county and is the same as an unlimited cull capacity. Although this is not a realistic scenario, it is important to demonstrate why cull capacities may be important to include in disease models such as this one. Ignoring resource capacities in models can be a logical fallacy in some cases by over-estimating the capabilities of disease management. Livestock culling was stopped after reaching the target density, so smaller target densities mean that a greater percentage of farms were culled compared with greater target densities for a specific ring cull. For target densities greater than 0 farms/km2, the largest farms were culled first. For each simulation, there was a daily limit to the number of farms that could be culled (cull capacity), representing resource limitations during an outbreak. Finally, the cull radius represents the radius from the IP that determines which farms are culled. For each simulation, one randomly selected farm was seeded with infection. Each unique control scenario was simulated 1,000 times. For each simulation, we recorded the number of cattle and sheep culled due to infection and ring culling, the number of farms culled due to infection and ring culling, and the length of the epidemic.

Epidemic impact measures

The epidemic impact measures (i.e., culled animals, culled farms, and epidemic length) were analyzed using SAS® PROC GENMOD (SAS/STAT 14.1; SAS Institute, Cary, NC, USA) to determine the impacts of the main effects (i.e., target density, cull capacity, cull radius). The data were subsetted by county and a generalized linear model (GLM) was fit to each of the three epidemic impact measures. Next, the epidemic impact measures (i.e., culled animals, culled farms, and epidemic length) were analyzed by fitting linear and quadratic regressions for each of three independent variables (i.e., target density after culling, cull capacity, cull radius) by county (3 × 2 × 3 × 4 = 36 regressions). Culled animals and farms refer to animals and farms that were culled because they resided in or were identified as IPs in addition to those that were within rings identified for culling. To determine the relative effects of target density, cull capacity, and cull radius on the most severe epidemics, we isolated the top ten percent of outbreaks for analysis (based on the top ten percent for each of culled animals, culled farms, and epidemic length) as well as the entire set of simulated outbreaks.

Linear and quadratic regressions were fit to data from each combination of the three independent variables (target density, cull capacity, and cull radius) and the three epidemic impact measures (culled animals, culled farms, and epidemic length) for Aberdeenshire, Cumbria, Devon, and North Yorkshire (Table S1). For each county, we identified the unique control strategy with the lowest epidemic impact measures across each of the three measurements (Table 2).

Table 2 Target density strategies minimizing cumulative epidemic impact measures by county and daily cull capacity for simulated FMD epidemics in four counties of the UK.

	Epidemic impact measure of interest	
	Minimum animals culled	Minimum farms culled	Minimum epidemic duration (days)	
Cull capacity	Value	Density	Radius	Value	Density	Radius	Value	Density	Radius	
Aberdeenshire	
5	591	0.4	5	1.94	0.4	5	18	0.05	3	
10	650	0.4	5	2.18	0.4	4	18	0.1	3	
20	635	0	0	2.08	0	0	18	0.05	4	
100	655	0.4	5	2.11	0.4	5	18	0	4	
Cumbria	
5	25,522	0.05	3	45	0.15	4	29	0.05	5	
10	24,982	0.15	3	46	0.15	3	30	0.05	4	
20	21,939	0.15	2	44	0.2	3	27	0.05	5	
100	21,617	0.15	2	42	0.2	4	29	0	5	
Devon	
5	5,293	0.15	0.5	17	0.15	0.5	23	0.05	5	
10	4,925	0.1	0.5	15	0.1	0.5	24	0.05	5	
20	4,844	0.1	0.1	17	0.1	1	24	0.05	5	
100	5,090	0.15	0.5	16	0.15	0.5	23	0	5	
North Yorkshire	
5	5,247	0.1	1	11	0.2	0.5	23	0.05	4	
10	5,252	0.2	0.5	12	0.15	0.5	22	0.05	5	
20	4,946	0.05	1	12	0.05	1	22	0.05	5	
100	4,424	0.15	0.5	10	0.15	0.5	21	0	5	
Note:

For each county and daily cull capacity (farms/day), we report the minimum average epidemic impact measure for each of the three epidemic impact measures of interest (total animals culled, total farms culled, and epidemic duration). We also report the corresponding target density and radius for each simulation scenario.

GLMs were used to assess the impact of the independent variables target density, cull capacity, and cull radius on the three epidemic impact measures (culled animals, culled farms, and epidemic length) for each county. A negative binomial distribution and log link function were used, and the Pearson chi-square and deviance values divided by the degrees of freedom were close to 1, which showed that the models were a good fit for the data (https://datadryad.org/stash/dataset/doi:10.5061/dryad.sbcc2fr6j). Target density, cull capacity, and cull radius were highly significant main effects by county, culled animals, culled farms, and epidemic length according to Type I and Type III analyses of likelihood ratios ( χ2, P < 0.0001). Since the Type III effects were significant, we looked at the differences and patterns within the levels of target density, cull capacity, and cull radius.

Sensitivity analysis

We assessed the sensitivity of cull strategy performance to variation in the spatial distribution of farms, the onset of preemptive culling, and the kernel shape parameter. Although the likelihood of a farm being infected depends on other factors (such as the size and species composition of the farm, production system characteristics, and on-farm biosecurity measures), proximity to infected premises is one of the main determinants of risk (Bessell et al., 2010), leading to the selection of a spatial kernel-based mechanism for modeling inter-farm transmission. In many farming settings, local clustering of farms is common, but the degree of clustering and the overall density vary considerably (Tildesley et al., 2009). To assess the sensitivity of the relative performance of each cull strategy, we compared model output where the control strategies were simulated on the true Cumbria farm locations and randomized Cumbria farm locations.

We also examined the sensitivity of disease spread to the onset of culling. We evaluated two cull timings, 14 and 30 days after the focal farm was infected. These cull timings are representative of a scenario in which disease managers are readily prepared and rapidly deploy resources (i.e., 14 days) and a scenario in which managers deploy resources at a normally expected rate (i.e., 30 days). All culls occurred on a single day at either 14 or 30 days after the focal farm was infected regardless of the number or species of animals on the culled farm or of different production systems (e.g., a confined facility such as a dairy vs an entirely pasture operation).

In order to evaluate the influence of model parameters upon the impact of these target-based culling strategies, we examined kernel shape parameter values ranging from 2 to 2.8, in 0.2 increments. Lower values of the kernel shape parameter correspond with a greater probability of transmission over long distances.

Results

From now on, we refer to the group of culled animals, culled farms, and epidemic length as the epidemic impact measures. The trends between culled cattle and culled sheep were similar, so we will present culled animals as a whole.

Epidemic impact analysis

The optimal control strategy varied by county and most of the time by the epidemic impact measure considered (Table 2). The only cases where a total ring depopulation (0 farms/km2) were optimal was for Devon and North Yorkshire when using epidemic length as a measure of impact. The simulation results differ among Aberdeenshire, Cumbria, Devon, and North Yorkshire so we will describe them separately. The SAS program script output can be found in (https://datadryad.org/stash/dataset/doi:10.5061/dryad.sbcc2fr6j). We then isolated the top ten percent of epidemics (based on the top ten percent for each of culled animals, culled farms, and epidemic length) to examine the simulated worst-case scenarios.

Aberdeenshire simulations

For Aberdeenshire, the independent variable target density (number of farms/km2 with live animals post-culling) has relatively small-magnitude effects on epidemic impact measures compared to the other three counties (Fig. 3). In terms of identifying the strategy with the fewest animals and farms culled, IP-only culling (target density of 0 farms/km2) and culling to 0.4 farms/km2 in a 5 km cull radius ring performed similarly with less than 700 total animals culled and around 2 farms culled. This pattern is consistent across all levels of cull capacity. When considering epidemic length, different levels of target density lead to small differences in epidemic length. Across all cull capacities, IP-only culling has the longest epidemics at 21 days whilst the shortest epidemic is around 18 days. The strategy with the overall shortest epidemic varies across cull capacity is 0.05 farms/km2 in a 5 km cull radius ring when cull capacity is 5 farms/day and is 0 farms/km2 in a 5 km cull radius ring when cull capacity is 100 farms/day. In Aberdeenshire, 33% of outbreaks had more than 1 farm culled (culled farms > 1) (Table S3). When isolating the upper ten percent of outbreaks, the differences in control strategies in terms of epidemic impact measures are magnified. Mid-range target densities appear to have mid-range levels of culled animals, culled farms, and epidemic lengths (Figs. S1–S3). These patterns are similar for both cull capacity and cull radius. As target density increases, culled animals and farms decrease while epidemic length increases. Therefore, the target density strategy may be beneficial in Aberdeenshire if reducing the number of culled animals and farms is more important than the length of the epidemic.

Figure 3 Effects of target farm density, daily farm cull capacity, and cull radius on culled animals, culled farms, and epidemic length for simulated FMD epidemics in Aberdeenshire, United Kingdom.

For panels (A–C), we present cumulative outbreak measures across cull radius (km) and target farm density (farms/km2) for different levels of daily cull capacity (farms/day). A target farm density of 0 farms/km2 represents total ring depopulation whilst 0.4 farms/km2 represents culling the fewest farms in each ring. Different marker styles and colors represent different target farm densities. We show six levels of target farm density (0, 0.05, 0.1, 0.15, 0.2, 0.4 farms/km2), four levels of daily cull capacity (5, 10, 20, 100 farms/day) and seven levels of cull radius (0, 0.5, 1, 2, 3, 4, 5 km) for a total of 168 stochastic simulation scenarios for Aberdeenshire. Each scenario was repeated 1,000 times. (A) Total animals (cattle + sheep) culled in thousands for each scenario where each point represents the mean from 1,000 simulations ± standard error. (B) Total farms culled for each scenario where each point represents the mean from 1,000 simulations ± standard error. (C) Epidemic length in days for each scenario where each point represents the mean from 1,000 simulations ± standard error.

Cumbria simulations

Compared with the three other counties, Cumbria consistently had the largest epidemic impact measures in terms of average culled animals, culled farms, and epidemic length. In Cumbria, more outbreaks spread beyond the initially infected premises compared with Aberdeenshire, Devon, and North Yorkshire. A mid-range density culling strategy of 0.05–0.2 farms/km2 and 2–5 km cull radius results in the fewest cumulative animals and farms culled (less than 30,000 animals and 80 farms) as well as the shortest epidemic length (less than 50 days) across all daily cull capacities (Fig. 4). As cull capacity increases, however, total ring depopulation (target density of 0 farms/km2) approaches the epidemic impacts from the previously mentioned strategies (Fig. 4). For the highest target density of 0.4 farms/km2, epidemic impacts increase as cull radius increases (Fig. 4). These same patterns hold when isolating the top ten percent of outbreaks (Fig. S4). For the cull radius, however, there appears to be a clearer indication that 2–3 km radii are most effective at reducing epidemic impacts (Fig. S4). Cumbria had the most outbreaks that spread beyond the initially infected premises, with 59.4% of 185,000 simulated outbreaks spreading beyond the primary case (culled farms > 1) (Table S3).

Figure 4 Effects of target farm density, daily farm cull capacity, and cull radius on culled animals, culled farms, and epidemic length for simulated FMD epidemics in Cumbria, UK.

For panels (A–C), we present cumulative outbreak measures across cull radius (km) and target farm density (farms/km2) for different levels of daily cull capacity (farms/day). A target farm density of 0 farms/km2 represents total ring depopulation whilst 0.4 farms/km2 represents culling the fewest farms in each ring. Different marker styles and colors represent different target farm densities. We show six levels of target farm density (0, 0.05, 0.1, 0.15, 0.2, 0.4 farms/km2), four levels of daily cull capacity (5, 10, 20, 100 farms/day) and seven levels of cull radius (0, 0.5, 1, 2, 3, 4, 5 km) for a total of 168 stochastic simulation scenarios for Cumbria. Each scenario was repeated 1,000 times. (A) Total animals culled in thousands for each scenario where each point represents the mean from 1,000 simulations ± standard error. (B) Total farms culled for each scenario where each point represents the mean from 1,000 simulations ± standard error. (C) Epidemic length in days for each scenario where each point represents the mean from 1,000 simulations ± standard error.

Devon simulations

Similar to Cumbria, the target density strategy leads to fewer cumulative animals and farms culled (5,000 to 20,000 animals and 17–35 farms) and shorter epidemics (20–50 days) at low daily cull capacities compared to total ring depopulation (17,000–37,000 animals, 45–85 farms and 50–75 days) (Fig. 5). The magnitudes of the differences between the lowest (0 farms/km2) and the highest (0.4 farms/km2) target densities are less than those for Cumbria, but these differences are greater than those seen in Aberdeenshire. Cull radii between 1 and 3 km were the lowest in average epidemic impact measures compared with the other levels of radii simulated. The same patterns hold when isolating the top ten percent of outbreaks and the magnitudes of the epidemic impact measures are much larger along with the differences between smallest and largest target density, cull capacity, and cull radius (Fig. S4). In Devon, there were 49.3% of 185,000 simulated outbreaks in which disease spread beyond the primary case (culled farms > 1) (Table S3).

Figure 5 Effects of target farm density, daily farm cull capacity, and cull radius on culled animals, culled farms, and epidemic length for simulated FMD epidemics in Devon, UK.

For panels (A–C), we present cumulative outbreak measures across cull radius (km) and target farm density (farms/km2) for different levels of daily cull capacity (farms/day). A target farm density of 0 farms/km2 represents total ring depopulation whilst 0.4 farms/km2 represents culling the fewest farms in each ring. Different marker styles and colors represent different target farm densities. We show six levels of target farm density (0, 0.05, 0.1, 0.15, 0.2, 0.4 farms/km2), four levels of daily cull capacity (5, 10, 20, 100 farms/day) and seven levels of cull radius (0, 0.5, 1, 2, 3, 4, 5 km) for a total of 168 stochastic simulation scenarios for Devon. Each scenario was repeated 1,000 times. (A) Total animals culled in thousands for each scenario where each point represents the mean from 1,000 simulations ± standard error. (B) Total farms culled for each scenario where each point represents the mean from 1,000 simulations ± standard error. (C) Epidemic length in days for each scenario where each point represents the mean from 1,000 simulations ± standard error.

North Yorkshire simulations

Similar to the other counties, the main effects of target density, cull capacity, and cull radius, were all highly significant (χ2, P < 0.0001). Across all epidemic impact measures and main effects, the patterns for North Yorkshire are highly similar to those found in the Devon simulations, with the magnitudes of epidemic impact measures slightly less in North Yorkshire compared with Devon (Figs. 5 and 6). A mid-range density culling strategy of 0.05–0.2 farms/km2 and 1–5 km cull radius results in the fewest cumulative animals and farms culled (less than 12,000 animals and 25 farms) as well as the shortest epidemic duration (less than 32 days) across all daily cull capacities (Fig. 6). As cull capacity increases, however, total ring depopulation (target density of 0 farms/km2) approaches the epidemic impacts from the previously mentioned strategies (Fig. 6). For the highest target density of 0.4 farms/km2, epidemic impact measures increase as cull radius increases (Fig. 6). As expected, when isolating the most severe outbreaks (the top ten percent) there is a more pronounced decrease in epidemic severity as cull capacity increases (Fig. S4). Similar to Devon, cull radii between 1 and 3 km are lower in terms of epidemic impact measures for both the entire data set and the most severe outbreaks (Fig. S4). North Yorkshire had a similar percentage of outbreaks that spread beyond the initially infected premises compared similar to Devon, with 48.58% of 185,000 simulated outbreaks spreading beyond the primary case (culled farms > 1) (Tables S3).

Figure 6 Effects of target farm density, daily farm cull capacity, and cull radius on culled animals, culled farms, and epidemic length for simulated FMD epidemics in North Yorkshire, UK.

For panels (A–C), we present cumulative outbreak measures across cull radius (km) and target farm density (farms/km2) for different levels of daily cull capacity (farms/day). A target farm density of 0 farms/km2 represents total ring depopulation whilst 0.4 farms/km2 represents culling the fewest farms in each ring. Different marker styles and colors represent different target farm densities. We show six levels of target farm density (0, 0.05, 0.1, 0.15, 0.2, 0.4 farms/km2), four levels of daily cull capacity (5, 10, 20, 100 farms/day) and seven levels of cull radius (0, 0.5, 1, 2, 3, 4, 5 km) for a total of 168 stochastic simulation scenarios for North Yorkshire. Each scenario was repeated 1,000 times. (A) Total animals culled in thousands for each scenario where each point represents the mean from 1,000 simulations ± standard error. (B) Total farms culled for each scenario where each point represents the mean from 1,000 simulations ± standard error. (C) Epidemic length in days for each scenario where each point represents the mean from 1,000 simulations ± standard error.

Sensitivity analyses

Farm size heterogeneities remained the same in the randomized location scenario as in the true scenario. Randomizing farm locations results in lower local densities of farms (Fig. S5; Tildesley et al., 2009), resulted in shorter epidemics, but did not change the relative performance of the target density and total ring depopulation strategies (Fig. S5).

We found that the relative performance of the culling strategies was unchanged between the two delays in initiation of preemptive culling we examined. We initially assumed it would take 30 days after the first infected farm was reported before enough resources would be mobilized to implement a preemptive culling campaign. We compared this with a scenario where policymakers were more prepared and could initiate preemptive control within 14 days of the first reported infection. When targeting the largest farms within control radii, the mean epidemic impact was decreased by approximately 270 farms and the mean epidemic length was decreased by 30 days, but the relative performance of the culling strategies was unchanged (Fig. S6).

We found that the relative performance of total ring depopulation and target density cull strategy was sensitive to variation in the patterns of pathogen dispersal in Cumbria (Fig. S7) as compared to the shape parameter of 2.2 used in the simulations described above. When the shape parameter was 2, we found the total ring depopulation policy often resulted in fewer losses of farms and shorter epidemics than did the target density policies. When the kernel shape parameter was intermediate (2.2–2.4), there was a small reduction in the number of farms culled compared with the total ring depopulation. When the shape parameter was in the higher ranges we examined (2.6–2.8), the average distance of transmission events was smallest, and we saw no distinct differences among any of the control policies because epidemics were generally smaller and easier to contain in these scenarios.

Discussion

The goal of our project was to determine if control of FMD under the conditions of the 2001 UK outbreak could be optimized by culling only to the level required to attain a threshold host density that would reduce disease spread and minimize epidemic impacts. Target density was an effective strategy in reducing overculling for simulations of some counties in the UK, especially Cumbria, Devon, and North Yorkshire. In these counties, the target density strategies with mid-range target densities (0.05–0.2 farms/km2) were particularly effective at low cull capacities compared to total ring population (Table 2; Figs. 4–6). This decrease in cumulative animals and farms culled and epidemic duration can likely be attributed to reduction of overculling while still reducing disease transmission by utilizing the target density strategy. For the same counties, Cumbria, Devon and North Yorkshire, we saw an increase in total animals and farms culled and epidemic duration at a target density of 0.4 farms/km2, which suggests that this level of target density was no longer effective at curbing transmission whilst reducing the amount of overculling in each ring cull (Figs. 4–6).

Cumbria consistently had the largest epidemics (Fig. 4), likely because it had the greatest average animal density (Table 1). While Devon has the greatest cattle density of the four counties (Table 1), which has been shown to be a greater driver of FMD epidemics compared with farm density (Meadows et al., 2018), Cumbria has nearly twice Devon’s mean sheep number per farm. We found that there are certain cases where target farm density culling may be more beneficial compared with total ring depopulation such as in Cumbria, which has higher mean farm density compared with Aberdeenshire and North Yorkshire (Table 1, Fig. 2), however, we found the benefits of the target density method were dependent on pathogen dispersal patterns (e.g., kernel shape). When we allowed pathogen dispersal to occur over greater distances (i.e., when the dispersal kernel was 2), the benefit no longer existed (Fig. S7). This behavior is expected since dispersal over greater distances could mean that the remaining farms in the control zone are not isolated enough to be protected from infection. This suggests the dispersal characteristics of the pathogen should be well understood before considering a target density strategy.

In Aberdeenshire, the target density strategy is more effective than the total ring depopulation strategy when using culled animals and culled farms as measures of epidemic impact, but the total ring depopulation strategy is more effective when considering epidemic length. The farm demography measures that appear to contribute most to the magnitude of the epidemics include cattle density, farm density, and total number of farms. To tease apart the contributions of each measure (Meadows et al., 2018), more simulated outbreaks should be considered in additional counties.

In most cases, mean epidemic impacts across counties decreased as daily farm cull capacity increased. We expect to see this result since more resources and control generally led to reduced epidemic severity. The departures we see from this pattern in Aberdeenshire (e.g., Fig. 3) could be due to epidemics being very mild in this county. Therefore, more animals and farms are culled than are being saved by controlling the disease. For Cumbria, similar to target farm density, optimal cull radii were identified (Fig. 4). Radii of 2–3 km were optimal because they have lower epidemic impact measures compared with the other simulated control strategies. We would normally expect the epidemic length to increase as culled animals and culled farms decrease, but this is not the case across all impact measures for Cumbria. For the other counties, the pattern of longer epidemics with decreased culling generally proceeds as we expect and is especially clear in Aberdeenshire (Fig. 3).

We observe that, while Devon has more farms (10,656) compared with Cumbria (7,884), Cumbria had higher mean epidemic impacts in all measures considered (Figs. 4 and 5), as was the case in the UK 2001 FMD epidemic. This may be due to Devon having fewer animals per farm and animals per km2 compared with Cumbria (Table 1). Devon and North Yorkshire were most similar in terms of mean epidemic impact measures (Figs. 5 and 6) and these patterns are likely driven by their similarity in average animal density (Table 1). For Aberdeenshire, the driving causes of the relatively low epidemic impact measures and the low number of outbreaks with disease spread are likely two-fold: (1) lower farm density and (2) lower animal density (Table 1). Since Aberdeenshire had the lowest number of outbreaks that spread beyond the primary case, this may suggest that outbreaks beginning in Aberdeenshire are less likely to spread compared with the other counties because of its differences in farm spatial pattern and demography. These results are consistent with the current literature on FMD, that farm demography measures such as livestock type (i.e., cattle, sheep), animal numbers (Ferguson, Donnelly & Anderson, 2001b), and mean farm density (Tildesley et al., 2010) explain regional variation in FMD transmissibility. Randomization resulted in smaller epidemics, but the effect of target density was similar. In FMD, spatial clustering has been found to be a driver of disease spread, and lack thereof can lead to smaller epidemics (Tildesley et al., 2010).

FMD can spread very quickly in some circumstances and ring culling is sometimes used to contain outbreaks. However, we have shown that reducing farm density through culling to reach a threshold density (i.e., target density strategy) might prevent overculling in some counties in the United Kingdom. We have shown that a concept similar to human social distancing can be applied to reduce epidemic impacts in FMD. Further systems to consider for applying target density culling, or a similar reduced-host-density strategy, would be systems that require host removal or culling to reduce host density as opposed to social distancing. Host removal or culling can come with great economic, sociological, ethical, and environmental costs. To avoid removing too many hosts from the systems, target density culling can be used in place of ring culling. Target density culling can be applied to agricultural and forest diseases such as wheat stripe rust (te Beest et al., 2011) and sudden oak death (Rizzo & Garbelotto, 2003), as well as vaccination strategies in FMD (Keeling et al., 2003).

While we found differences in epidemic impact measures across culling strategies and counties, the counties were closed systems, meaning that spread of inoculum among counties is not considered. The target density strategy requires determining which farms to cull beyond each IP. For this study, we prioritized the culling of the largest farms. In other words, for each ring, we sorted farms by the number of livestock, and we culled farms with the largest number until we reached the target farm density within the ring. Selecting the largest farms for culling may not be appropriate in all cases. Other strategies may include but are not limited to, culling random farms, farms with the highest risk of transmission, and most connected farms (VanderWaal et al., 2016). These strategies might also be affected by animal species, production system, and IP density. In the UK, it would be important to determine the effect of each strategy by county in terms of epidemic impact measures and to consider the political, economic, and social ramifications of each strategy (Mort et al., 2005; Probert et al., 2016). Some selection strategies may lead to lower epidemic impacts, while others may target large-scale farms. The intersection of epidemic impact and farm size is an important discussion that would require further studies and discussions with policymakers and farmers in each area of concern.

The current minimum control strategy to combat FMD in the European Union is to cull IPs and dangerous contacts (DCs) (Department for Environment, Food and Rural Affairs of the United Kingdom, 2011). Now that we have shown that IP culling along with the target density strategy can be effective for some counties in the UK, further simulations with the addition of DC culling would be beneficial to capture the current minimum control strategies. In other words, we can apply the target density strategy along with DC culling in our simulations to determine the effectiveness of applying target density strategies in addition to the current minimum control.

We isolated the top ten percent of epidemics to examine the worst-case scenarios. Disease managers are often interested in the spatiotemporal spread of high-risk and superspreading events (Lau et al., 2017; Moreno et al., 2020). In this case of simulating the spread of FMD in the UK, the patterns of disease spread hold similar when only the top ten percent of outbreaks were considered. This may not be the case with other long-distance dispersal diseases. It is essential to isolate the ‘worst-case scenarios’ for a given disease system to determine whether there are differences in relative spatiotemporal spread, and, if so, to determine the main drivers of these patterns in disease severity.

For long-distance dispersal systems, it is essential to determine whether or not target density strategies are feasible. In terms of whether a target density approach could be legally implemented, it is worth noting in the UK that livestock farms were culled for many reasons during the 2001 FMD outbreak. Dangerous contact culling was targeted at farms that were specifically designated to be high risk, either owing to known contact with an infected premises or owing to a perceived increased risk of exposure. Additionally, contiguous culling targeted farms that bordered infected farms and finally, ring culling was implemented in some regions (specifically Cumbria and Dumfries and Galloway). However, not all farms within a ring were targeted and the cull was predominantly targeted at sheep farms in those counties. There is therefore a clear precedent for culling strategies that identify farms based upon risk and for ring culling that does not target all farms. A target density approach would combine many aspects of dangerous contact culling with fixed radius ring culling in order to identify farms within a ring that may reduce the risk of significant spread from the source farm. The legality of target-density-based strategies should be thoroughly considered before considering this as a potential control strategy in disease management for other long-distance dispersal systems.

We have found that a target density culling strategy, compared with a total ring depopulation policy, may reduce FMD epidemic impacts in some counties in the UK, but the optimal cull strategy in terms of target density, cull capacity, and cull radius depends on farm demography, spatial pattern, and pathogen dispersal characteristics. To prevent overculling and the associated economic, ethical, and psychological impacts (Mort et al., 2005; Peck, 2005), the target density strategy may be worth considering in decision-making processes for future FMD control. In addition, organizational stress associated with depopulation can result in relaxation of important hygiene practices used for FMD control, and the target density approach has the potential to alleviate organizational stress by reducing the number of farms that need to be depopulated. For plant systems, ring culling is a common strategy used to combat disease (Parnell et al., 2009). Risk-based culling has been found to be more effective than ring culling (Hyatt-Twynam et al., 2017; te Beest et al., 2011). However, a target—density-based control strategy may also lead to lower epidemic impacts compared with total ring depopulation and could be worth exploring as an alternative control strategy to risk-based culling as exemplified by our control strategy comparison in FMD.

Supplemental Information

Supplemental Information 1 Effects of target density, daily farm cull capacity, and cull radius on culled animals, cattle only, and sheep only for foot-and-mouth disease in four UK counties based on all 1000 simulations.

The four counties are shown in panels of different color. Target size culling implementation variables are on the x-axes and include target farm cull density, daily farm cull capacity, and cull radius. Farm response variables are on the y-axes and include total culled animals (cattle and sheep) per county, total culled cattle per county, and total culled sheep per county. The points and error bars indicate means ± standard errors. The curves are quadratic regressions fit to the means and shaded areas are standard errors of the regressions. (A–) The y-axis is log10-transformed; (D–F) The x-axis is log10-transformed. Each data point is the mean over 1,000 simulations.

Supplemental Information 2 Effects of target density, daily farm cull capacity, and cull radius on culled animals, culled farms, and epidemic length for FMD in four UK counties based on all 1000 simulations.

The four counties are shown in panels of different color. Target size culling implementation variables are on the x-axes and include target farm cull density, daily farm cull capacity, and cull radius. Farm response variables are on the y-axes and include total culled animals (cattle and sheep) per county, total culled cattle farms per county, and mean epidemic length per county. The points and error bars indicate means ± standard errors. The curves are quadratic regressions fit to the means and shaded areas are standard errors of the regressions. (A–I) The y-axis is log10-transformed; (D–F) The x-axis is log10-transformed. Each data point is the mean over 1,000 simulations.

Supplemental Information 3 Effects of target density, daily farm cull capacity, and cull radius on culled animals, culled farms, and epidemic length for FMD in four UK counties based on the 10% most severe simulations.

The four counties are shown in panels of different color. Target size culling implementation variables are on the x-axes and include target farm cull density, daily farm cull capacity, and cull radius. Farm response variables are on the y-axes and include total culled animals (cattle and sheep) per county, total culled cattle farms per county, and mean epidemic length per county. The points and error bars indicate means ± standard errors. The curves are quadratic regressions fit to the means and shaded areas are standard errors of the regressions. (A–I) The y-axis is log10-transformed; (D–F) The x-axis is log10-transformed. Each data point is the mean over the 10% most severe epidemics for each epidemic impact (culled animals, culled farms, and epidemic length.

Supplemental Information 4 Effects of target density, daily farm cull capacity, and cull radius on culled animals, culled farms, and epidemic length for FMD in four UK counties based on the 10% most severe simulations.

(A) target farm cull density and culled animals, (B) target farm cull density and culled farms, (C) target farm cull density and epidemic length, (D) daily farm cull capacity and culled animals, (E) daily farm cull capacity and culled farms, (F) daily farm cull capacity and epidemic length, (G) cull radius and culled animals, (H) cull radius and culled farms, (I) cull radius and epidemic length. The points indicate means over 100 simulations (the top 10 percent of each epidemic impact measure, i.e., culled animals, culled farms, and epidemic length. The curves and shaded regions are quadratic regressions fit to the data by county ± standard error. The different colors and shapes indicate different counties. (A–I) The y-axis is log10-transformed. (D–F) The x-axis is log10-transformed.

Supplemental Information 5 Simulated effect of the spatial extent of preemptive control methods on the length of foot-and-mouth disease epidemics of livestock in Cumbria, UK, assuming different spatial patterns of farms.

(A) FMD outbreaks were simulated using true and random Cumbria farm locations, indicated by points. (B) Lengths of FMD epidemics are shown for true (dashed lines) and random (solid lines) of Cumbria farm locations. Infected farms were located and culled to achieve the target farm density within the control radius. Each panel shows a unique combination of daily cull capacity (50 farms/day or unrestricted; columns) and prioritization for culling farms within the control radius (randomly select farms or target the largest farms; rows).

Supplemental Information 6 Relationship between the spatial extent of culling and the total epidemic impact and epidemic length when culling was initiated either randomly or by targeting the largest farms first in simulations.

Total epidemic impact includes number of farms lost to both infection and control methods, upper four panels) and epidemic length (lower four panels). Susceptible farms located within the control radius of an IP were culled to achieve the target farm density (color scale) within the control radius. Each panel shows a unique combination of daily cull capacity (50 farms/day or unrestricted; columns) and prioritization strategy for culling farms within the control radius (randomly select farms or target the largest farms; rows).

Supplemental Information 7 Relationship between the spatial extent of culling and the total number of farms culled (top row) and epidemic length (bottom row) as influenced by the dispersal kernel shape parameter.

Results shown are the mean epidemic impact (lines) and 95% C.I. (shaded regions) for each target density cull strategy (color scale). The value at 0 km control radius represents the ‘stamping-out’ control strategy. Each column of panels show results for the indicated value of the kernel shape parameter when the daily cull limit is 50 farms/day and the largest farms within control radii are prioritized for cull. Each data point is the mean over 1,000 simulations.

Supplemental Information 8 Summary statistics by county for effects of target farm density, daily farm cull capacity, and cull radius on culled cattle, culled sheep, total culled animals, culled farms, and epidemic length.

The statistics include number of simulations (N), mean, standard deviation (sd), standard error (se), and 95% confidence interval (ci).

Supplemental Information 9 Linear and quadratic regression parameters by county for effects of independent variables (first factor in subtitles) on dependent variables (second factor in subtitles) for simulations of FMD in the UK.

* = P < 0.05, ** = P < 0.01, and *** = P < 0.001.

Supplemental Information 10 Number of simulated foot-and-mouth disease outbreaks that spread to at least one farm from the seeded farm (culled farms > 1) as affected by cull radius and cull capacity in four UK counties.

Percentages of total simulations outbreaks are indicated in parentheses.

We acknowledge the contribution of Matt J. Keeling from the University of Warwick who developed the original version of the simulation-based model used for this study of controlling FMD using target-density-based strategies. Jean Bertrand Contina made contributions to analysis and interpretation of data. We also thank the CoSINe IT Services at Oregon State University for providing support and access to the Unix High Performance Computing Cluster for running our simulations.

Additional Information and Declarations

Competing Interests

Author Contributions

Data Availability

Amanda J. Meadows is employed by Ginkgo Bioworks.

Rachel L. Seibel conceived and designed the experiments, performed the experiments, analyzed the data, prepared figures and/or tables, authored or reviewed drafts of the article, and approved the final draft.

Amanda J. Meadows conceived and designed the experiments, performed the experiments, analyzed the data, prepared figures and/or tables, authored or reviewed drafts of the article, and approved the final draft.

Christopher Mundt conceived and designed the experiments, authored or reviewed drafts of the article, and approved the final draft.

Michael Tildesley conceived and designed the experiments, authored or reviewed drafts of the article, and approved the final draft.

The following information was supplied regarding data availability:

The simulation data and novel SAS code used for data analysis are p archived on Dryad: Seibel, Rachel et al. (2021). Statistical analysis code for output from a model used to simulate foot-and-mouth disease dynamics in the United Kingdom [Dataset]. Dryad. https://doi.org/10.5061/dryad.sbcc2fr6j.

This article also uses model code adapted from previously published studies and further information on these studies is included in the Methods.

The data on the 2001 UK FMD outbreak are available on request from the Department for Environment, Food, and Rural Affairs (DEFRA) of the government of the United Kingdom. Access to these data, including appropriate DEFRA contact information, is available at: https://www.gov.uk/government/organisations/department-for-environment-food-rural-affairs.

Additional data summaries are available in the Supplemental Files.

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
