# Peer review of "Modeling target-density-based cull strategies to contain foot-and-mouth disease outbreaks"

_PeerJ, doi:10.7717/peerj.16998_

## Round 0.1 · original submission · Major Revisions

Please address all reviewers' comments.

·

Basic reporting

This is an interesting analysis on killing of animals around an FMD outbreak in different counties in the UK. The audience for such a paper would be governmental risk managers. In my opinion the paper is insufficiently clear for this audience. E.g., the authors do not mention that reduction of the number of farms that are depopulated could be seen as an improvement from an ethical point of view. For governmental risk managers, ethics is one of the drivers for their decisions.

The authors clearly indicate that their analysis is based on financial and logistical constraints (line 44). But there is no economic analysis, how can it be based on financial constraints? In an economic analysis in the Netherlands, the duration of the outbreak was an important economic parameter.
Backer, J. A., Bergevoet, R. H. M., Hagenaars, T. J., Bondt, N., Nodelijk, G. A., van Wagenberg, C. P. A., van Roermund, H. J. W. (2009). Vaccination against foot-and-mouth disease differentiating strategies and their epidemiological and economic consequences. Wageningen, Wageningen-UR: 160 (https://library.wur.nl/WebQuery/wurpubs/386271).

A stand-still of movement of animals is the most important control tool. The Kernel represents the spread independent of animal movement. In most cases airborne transmission is negligible in FMD outbreaks, so the source of spread is the movement of people and inanimate objects between farms. Better hygiene could stop transmission, this is not discussed, but extremely important for a governmental risk manager.

The models implicitly assume that there is not transmission due to depopulating farms. But implementing ring killing increases the number of persons on farms and thereby the risk that these persons could move virus to other farms. Due to the high number of farms (often 3 to 4 more than the number of IPs) that must be depopulated, the stress on the organisation can lead to reduced implementation of hygienic measures (in the 1997 CSFV outbreak in the Netherlands it can be assumed that the overwhelming depopulation has contributed to spread of the disease), which can contribute to more transmission. In the model the killing of animals is assumed without risk (which is invalid). The authors should discuss the risk of additional spread due to killing all susceptible animals on neighbouring farms.

Killing healthy animals on neighbouring farms (in many cases less than 5% of the neighbouring farms are truly infected) has serious ethical consequences. Even though the authors limited themselves to financial and logistical constraints, these ethical consequences should be mentioned in the discussion.

Experimental design

Form the paper it is not clear how the models were run. There are a high number of possible combinations of radius for depopulation, target density and capacity. I just do not understand how I should read figure 4. If I select the best target density, for which radius and which capacity is this valid?

Validity of the findings

Based on the presentation it is very difficult to interpret the results and their validity

Additional comments

General comments:
Abstract:
I'm missing a specific reference to "Ethics". I assume the reduction in "overculling" is driven by ethics.

Introduction:
Line 69 - 71: The meaning of the sentence "During …… sheep" is not clear to me.
Line 80 - 91: EU directive 2003-85 was the first agreement that would allow vaccination to live. But since 2003 no FMD epidemics have occurred in which vaccination was considered. So, I do not believe the other limitations were relevant as there was no need to control FMD by vaccination in the EU.

Methods:
Line 135: Boender et al. (2010) use a different function, the risk with distance reduces less than in the function used by the authors. Are the different functions compared? Is there a reference to this comparison? What is the best function?
Boender, G. J., van Roermund, H. J. W., de Jong, M. C. M., Hagenaars, T. J. 2010. Transmission risks and control of foot-and-mouth disease in The Netherlands: Spatial patterns. Epidemics 2(1); 36-47.
Line 142, 144, 148 and 149: What kind of farms were culled? 2961 infected farms, 2961 neighbouring farms?
Line 157: Common? No most endemic countries implement vaccination to control disease (most of the countries in continental Europe implemented prophylactic vaccination, only Denmark and the UK used stamping out). In Africa, Asia and South-America ring culling was never implemented.
Line 158: "ring depopulation" or "ring culling" not both.
Line 174 - 175: What does the author mean with "Livestock".
Line 187 - 191 is a repetition of line 162 - 166, the wording is only slightly different. I suggest removing it in one of both places.
Line 220-232: How did you select the best GLM model?

Results:
Line 275: Do you mean type I and type III sum of squares in the ANOVA. Why did you not use AIC as a selection criterion?
Line 282: "For target density", what do you mean, what is the Result variable in the model and what are the explanatory variables?
Line 300: In the other counties it is indicated what percentage of simulations had culled farms >1. Why is this missing in Aberdeenshire?
Line 356: Indicate what the initial shape variable was! Indicate how the kernel changes when changing the shape parameter.
Line 358: Improvement in what? What was the target density? Very unclear.

Discussion:
Line 365-372: delete and just restate the objective.
Line 423: Delete "Highly". Based on NSP surveillance in endemic settings the R-naught is between 1.3 and 1.7 which is not very high.

Reviewer 2 ·

Basic reporting

no comment

Experimental design

no comment

Validity of the findings

no comment

Additional comments

Review report: Modeling target-density-based cull strategies to contain foot-and-mouth disease outbreaks


Introduction

The authors employed a previously developed stochastic simulation algorithm to assess the viability of target density-based culling as a potential alternative to total ring depopulation during an FMD outbreak in the UK. This research delves into an intriguing topic that could offer valuable insights into control strategies aimed at minimizing culling overshoot and reducing costs. In order to enhance the manuscript's quality, I have included several suggestions and comments below.

Comments

I would discourage the inclusion of citations in the Abstract. Instead of specifically mentioning "simulation algorithm originally developed by Keeling et al. 2001," the authors could state "simulation algorithm developed earlier" or "an existing simulation algorithm was used." This approach avoids unnecessary citation within the abstract
To enhance readability, it would be beneficial to avoid beginning two consecutive sentences with the same phrase. For example, in the Abstract, rephrasing lines 30 or 31 would contribute to a more varied sentence structure.
In the Abstract, when stating "We found that all of the cull strategy parameters were drivers of epidemic impact," it would be helpful to list the parameter names in parentheses immediately after this statement.
Providing justifications for the model assumptions would be valuable. For instance, in line 123, where the authors state "We assume no livestock movement between farms and the model does not allow for new hosts to enter the system," it would be reasonable to justify this assumption. A possible explanation could be the short time frame of the simulation which would support the assumption of no inter-farm movement of livestock.
Additionally, in line 356, where the word "better" is used, it would be beneficial to provide further explanation or context to clarify what "better" means, such as discussing specific aspects or criteria for comparison.


The description of the model used can be improved. While I acknowledge that this is an existing model employed in previous studies, it is still necessary to provide a clear explanation of the model. It would be beneficial to include, at the very least, a schematic diagram illustrating the model mechanism and, if possible, the model equations, or equations governing some of the processes in the simulation (This could be in the supplement) .

Furthermore, the criteria used for targeting farms for depopulation were not immediately apparent. Although briefly mentioned in the Discussion section, I believe it is crucial to discuss these criteria in the Methods section. Additionally, it would be helpful to describe the process for selecting animals within a farm for culling, such as whether it was done randomly or by another method.

The authors mentioned that the model was fitted to data. It would be advantageous to provide more details about the fitting procedure and how the confidence bounds (ribbons) were derived.

To make the discussion section more concise, it could be useful to present some of the county-specific results in a table.

Reviewer 3 ·

Basic reporting

Quality of English is good and the manuscript is easy to read. Nice job.

Experimental design

Well done on a thorough and methodical approach to the study. I think the manuscript presents a useful and interesting piece of modelling and will contribute to the already substantial body of work centered around the UK 2001 outbreak.

Validity of the findings

Covered under additional comments.

Additional comments

Abstract
- Line 10: Mike is designated the corresponding author on page 4 while Chris is designated the corresponding author on page 6.
- Line 14 and throughout: Consider using the word ‘outbreak’ instead of ‘epidemic’. Epidemics tend to signify long standing or widely spread. Outbreak is a more general term and I think better aligns with an incursion of exotic disease that triggers an emergency response. Similarly, I would consider using the term ‘disease spread’ rather than ‘epidemic spread’, and ‘outbreak impact’ instead of ‘epidemic impact’.
- Line 17: Saying that total ring depopulation is ‘often used’ for emergency animal disease outbreaks is quite a big statement. Can you confidently make it across a range of countries, diseases, and host densities? Mass slaughter of seemingly healthy animals due to either pre-emptive culling or vacc-to-die will trigger strong reactions/objections from stakeholders. Decision makers may be reluctant to trigger these measures, particularly after the UK 2001 experience. Perhaps soften your language to how it has been used in the past and remains a policy option under certain emergency situations.
- Line 23: In addition to over-culling, another motivation for partial culling inside the cull zone might be insufficient resources to conduct absolute culling in a timely manner.
- Line 32: The term ‘take off’ is quite vague and a tad informal. See related comment in Body.
- Overall: I find the abstract a bit vague and doesn’t do justice to the findings of the study. It’s best to have strong clear statements in your abstract to entice browsing readers to engage further with the paper.
Body:
- Line 43: Perhaps ‘deploy’ instead of ‘distribute’?
- Line 53: I don’t really see the value of including supporting examples from plant/environmental health. Your target audience will mainly be in animal health. Policy and techniques for surveillance and control differ markedly between the two domains. Stakeholders and the general public are likely to react/object far more strongly to pre-emptive culling of livestock than pre-emptive thinning of a plant disease host.
- Line 77: Culling of vaccinated animals may occur quite some time after vaccinating, rather than ‘followed by’. The timing will depend on policy, how well the outbreak is being contained, and resource availability.
- Line 120: A limitation of a state-transition model is that all animals on a farm become infectious at once. In reality, infectiousness increases over time as spread progresses within the herd, and thus the infectious pressure exerted on susceptible farms waxes and wanes rather than being a light switch. No action suggested – just fyi.
- Line 126: Perhaps mention that in this modelling approach all potential FMD spread mechanisms (direct, indirect & airborne) are aggregated into the transmission kernel.
- Line 151: Define ‘local farm density’. There is a reference to Figure 2 which uses the term ‘mean farm density’. Best to define terms early and stick to them rather than using synonyms.
- Line 157: Is it fair to say it is common in both endemic and exotic contexts? Perhaps soften the term ‘common’ to ‘available’.
- Throughout: Clean up the definition and usage of the terms ‘target farm density’, ‘target density’ and ‘density’. You use the term ‘target farm density’ twice prior to defining it on Line 162 as ‘target density’. You switch back and forth between ‘target farm density’ and ‘target density’ throughout the manuscript as well as redefining it on Line 254 as ‘density’. I suggest defining a single term ‘target density’ early in the manuscript and then sticking to it throughout.
- Line 191: You use the term ‘cull capacity’ prior to defining it here. Consider defining all your terms early in the manuscript.
- Line 200: I think you should provide a reason why pigs were not included in the study. For example, whilst pigs can play an important role in FMD outbreaks (reference), there were very few pig IPs in the UK 2001 outbreak. As such, they were omitted from the study to simplify….…
- Line 201: Perhaps mention that the simulations were run independently and separately in each county (as opposed to a more national-scale study where counties can interact epidemiologically).
- Line 209: Perhaps ‘over-estimating’ rather than ‘misrepresenting’.
- Line 213: Were culling activities stopped or queued when the daily capacity was reached? It’s unclear whether each simulation day has a clean slate with respect to culling requirements & activity, or whether culling jobs are queued and potentially processed the next day.
- Line 216: Did you consider investigating whether the nature of the seed farm (species, size, production system) influenced outbreak characteristics?
- Line 221: Why was number of IPs not included as a distinct outbreak outcome?
- Line 231 and 280: What was the severity criteria for the top 10%? Number of IPs? Outbreak duration? Number of culled farms? Number of culled animals?
- Line 236: Also contributing are production system characteristics and on-farm biosecurity measures.
- Line 238: after (Bessell et al., 2010)…..leading to the selection of a spatial kernel-based mechanism for modelling intra-farm transmission.
- Line 247: Does the model allow the size of the farm to influence the time taken to cull? What about production system and species? Culling a 50-head dairy farm is quite a different proposition to a 1000-head pastoral beef farm.
- Line 253: Consider moving descriptions of methodology (e.g., regression techniques) from Results to the Method section.
- Line 254: Just define ‘target density’ in the intro and stick to a single term throughout. Same idea for ‘cull capacity’ (vs ‘daily farm cull capacity’) and ‘cull radius’ (vs ‘control radius’).
- Line 257: Standardise the terminology for outbreak outcomes. Decide on either ‘epidemic impact data’ (Line 221) or ‘Epidemic impact measures’ (Line 225). Define early. Define once. Be consistent in the usage.
- Lines 304, 314 and 339: ‘Took off’ is too vague a term. An outbreak can be assessed in a variety of ways, e.g., spread or not beyond the primary case, number of IPs, rate of increase of IPs over a fixed period, number of infected farms (which may differ from the number of IPs), infection area, number of infected animals, number of culled animals, cost, control duration, etc.
- Lines 315, 328, 340, 414: Do you mean primary case (where infection was seeded) or index case (where infection was first detected)?
- Line 365-370: Plant and aquatic disease references may have limited traction for an animal health audience. One Health cross-pollination is important, but best to stick to animal health terminology, concepts, and references for a niche paper like this. Also, there will generally be greater reticence by policy makers / general public to pre-emptively cull mammals over plants/fish – so the host density reduction analogies may not be well received.
- Line 372: Keep in mind there is significant underlying heterogeneity in FMD outbreaks. For example, species (cattle vs sheep vs goats vs pigs vs deer), domestic vs wild populations, production system (intensive vs extensive), on-farm biosecurity, marketing systems (network-driven direct transmission that varies regionally and seasonally vs spatial kernel-driven), environmental influences on indirect spread, environmental and species/production system influences on the potential for airborne spread, etc. It’s not necessarily as simple as just reducing host density.
- Line 370: Consider discussing how intra-farm transmission can be modelled as frequency dependent or density dependent and how this choice may influence the effect of reducing host density on outbreaks.
- Line 426: I doubt animal health readers will relate to the comparison of pre-emptively culling livestock with Covid social distancing. Livestock owners, veterinarians, response crews were devastated by the 2001 response (see Mort et al., 2005; Peck 2005). It might be nice to mention the social benefit of reducing the numbers of culled farms/animals should a ring culling policy once again be pursued.
- Line 438-443: Other considerations for setting culling priorities might species, production system, IP density.

Mort, M., Convery, I., Baxter, J., & Bailey, C. (2005). Psychosocial effects of the 2001 UK foot and mouth disease epidemic in a rural population: qualitative diary-based study. BMJ (Clinical research ed.), 331(7527), 1234. https://doi.org/10.1136/bmj.38603.375856.68
Peck, D. (2005). Foot and mouth outbreak: Lessons for mental health services. Advances in Psychiatric Treatment, 11(4), 270-276. doi:10.1192/apt.11.4.27

---

## Round 0.2 · Minor Revisions

Please, address the minor revisions requested by reviewer #1

·

Basic reporting

The report is not concise, but reads wel.

The abstract is quite long. I was taught that the abstract should contain 4 sentences (Why, How, What and So what). Nowadays concise writing is difficult as copy paste is so easy, when people were using typewriters, that was different.

Experimental design

This is okay.

Validity of the findings

This is okay.

Additional comments

Line 48: According to WOAH (the standard for disease control) OUTBREAK means the occurrence of one or more cases in an epidemiological unit. For a group of epidemiological related outbreaks they use the term event. So, I think the term "outbreaks" is used correctly, although it could be replaced by "epidemic" (as more specific term compared to the WOAH term event).

Line 57: I assume you mean "stamping out animals on infected premises" when you say "reactive culling", but I'm not sure. Be precise and concise.

Line 60: "preemptive control measures" is quite broad. Do you mean a "stand still of movement of animals" or "screening around infective premises" or "preemptive killing of animals on neighboring farms". Be precise and concise.

Line 95: Is "small density" correct English, I would think that density is high or low, and farms would small or large (independent of the area they cover). So, do you mean a low density of small farms, or just a low density of farms. Be precise and concise.

Line 118: "As FMD spreads rapidly with farms" do you mean farms move quickly and FMD moves with them? Or do you mean "As FMD spreads rapidly within farms"? Be precise and concise.

Line 194: Where do the number of cattle come from, they are not given in the example, line 189 it is said "varying numbers of cattle" but no specific number. We would not consider 4 farms holding 10 cattle, commercial holdings.

Line 223: it should probably read (0.0, 0.05, 0.1, 0.15, 0.2 and 0.4 farms/km2)

Line 356: 91279 out of 185000 would have a 95% CI (Clopper–Pearson interval) of 49.1 to 49.6, so 3 significant digits is sufficient in this case 49.3%

Table S3 in the last column the number between brackets is the percentage I think. Please reduce the number of digits (3 significant digits is sufficient) and add % and text explaining what the reader sees in the table.

Line 463: FMD is not a highly contagious disease, in endemic settings the R-naught is 1.3 – 1.8. Furthermore contagiousness is not determined by the disease agent, but often largely by the people handling the infected animals, animal density, contact structure. So, rephrase, e.g. "FMD can spread very quickly in some circumstances".

Figure 2 is missing lower panel and inset. I do not understand this graph, how can average density decrease with radius? Assuming an evenly distributed area, the line would be horizontal. Now the author is considering a non-evenly distributed area, then for some farms in a densely populated area the density would go down, but for farms in a sparsely populated area the density could go up. But the average would probably still be a horizontal line.

I'm lacking sufficient information in the table headings of the supplementary tables, and also in the supplementary figures.

Reviewer 2 ·

Basic reporting

no comment

Experimental design

no comment

Validity of the findings

no comment

Additional comments

The authors have sufficiently addressed all my comments.

Reviewer 3 ·

Basic reporting

No comment - covered in initial review.

Experimental design

No comment - covered in initial review.

Validity of the findings

No comment - covered in initial review.

---

## Round 0.3 · accepted · Accept

Thank you for addressing the reviewers' comments. Your manuscript is now deemed ready for publication.